# Expression of p53, p63, p16, Ki67, Cyclin D, Bcl-2, and CD31 Markers in Actinic Keratosis, In Situ Squamous Cell Carcinoma and Normal Sun-Exposed Skin of Elderly Patients

**DOI:** 10.3390/jcm12237291

**Published:** 2023-11-24

**Authors:** Alise Balcere, Māris Sperga, Ingrīda Čēma, Gunārs Lauskis, Maksims Zolovs, Māra Rone Kupfere, Angelika Krūmiņa

**Affiliations:** 1Department of Dermatology and Venereology, Riga Stradiņš University, LV-1010 Riga, Latvia; mara.rone-kupfere@rsu.lv; 2Department of Pathology of Infectious Diseases, Riga East University Hospital, LV-1006 Riga, Latvia; maris.sperga@rsu.lv; 3Department of Maxillofacial Surgery and Oral Medicine, Riga Stradiņš University, LV-1007 Riga, Latvia; ingrida.cema@rsu.lv (I.Č.); gunars.lauskis@rsu.lv (G.L.); 4Statistics Unit, Riga Stradiņš University, LV-1007 Riga, Latvia; maksims.zolovs@rsu.lv; 5Institute of Life Sciences and Technology, Daugavpils University, LV-5401 Daugavpils, Latvia; 6Department of Infectiology, Riga Stradiņš University, LV-1007 Riga, Latvia; angelika.krumina@rsu.lv

**Keywords:** age, UV damage, immunohistochemistry, cell cycle regulatory markers, skin cancer

## Abstract

Background: Age and cumulative exposure to ultraviolet (UV) light are primary contributors to skin cancer development. Regulatory proteins within the cell cycle are essential for the homeostasis of squamous epithelium. Methods: This study assessed the expression of immunohistochemical markers p53, p63, p16, Ki67, Cyclin D, Bcl-2, and CD31 in keratinocyte intraepithelial neoplasia (actinic keratosis and squamous cell carcinoma in situ) compared to normal skin. The objective was to distinguish disease-specific changes from those attributable to ageing and sun exposure in elderly skin. Results. Analysis included 22 actinic keratoses (AK), 7 in situ squamous cell carcinomas (SCC), and 8 normal skin biopsies. The mean age was 78.1 years for the AK/SCC group and 73.8 years for controls, with no significant age difference noted between the groups. The AK/SCC group exhibited a higher occurrence of amorphous masses, higher intensity of p53, lower Bcl-2 expression in the epidermis, higher Bcl-2 expression in the dermis, and higher CD31 expression in the dermis, all of which were statistically significant (*p* < 0.05). Conclusions: The study identifies distinct differences in the presence of amorphous masses and the expression levels of p53, Bcl-2, and CD31 between sun-exposed skin and in situ cutaneous squamous cell carcinomas, including actinic keratoses.

## 1. Introduction

Cumulative ultraviolet (UV) damage constitutes the principal environmental factor leading to keratinocyte intraepithelial neoplasia and the subsequent emergence of skin cancer. Actinic keratoses (AK), often classed as either precancerous lesions or squamous cell carcinomas (SCCs) in situ, are prevalent conditions in general and dermatological practice [1,2,3,4,5]. These lesions typically develop in older individuals with fair skin. Their prevalence demonstrably increases from the third decade of life and exceeds 90% in individuals beyond 80 years of age [5,6]. Morphologically, AK and SCC are widely regarded as existing on a continuous nosological spectrum but at distinct evolutionary stages [7]. Histopathologically, AK is distinguished by basal keratinocyte atypia, characterized by loss of polarization, cellular crowding, and overlapping. In more advanced stages, this atypia can encompass nearly the entire thickness of the epidermis. AK is stratified into three grades based on the extent of vertical involvement by atypical keratinocytes: Grade I involves the lower third, Grade II extends to the lower two-thirds, and Grade III affects more than two-thirds of the epidermis without reaching full-thickness [8]. The bowenoid subtype of AK, notable for its full-thickness atypia, complicates differentiation from SCC in situ. Dermatoscopically, bowenoid AK is marked by a uniform distribution of glomerular vessels and the lesion is found within a cancerization field. This contrasts with Bowen’s disease, a type of SCC in situ which displays a dermatoscopic pattern of irregularly distributed, clustered vessels, and is typically not associated with field cancerization [5]. Although bowenoid AK is typically described as exhibiting full-thickness dysplasia without adnexal involvement [9], Fernández-Figueras et al. [10] have observed that AK, even when confined to the lower third of the epidermis, may extend into adnexal structures. In this study, lesions with full-thickness involvement are unequivocally classified as SCC in situ.

The p53, p63, p16, Ki67, Cyclin D, and Bcl-2 proteins are essential regulators of the cell cycle [1,2,3,4,5,7], contributing not only to the development and homeostasis of squamous epithelium [11,12] but also to tumor genesis [13,14]. Their expression is modified by both age [15,16] and UV radiation exposure [12,17,18,19]. Specifically, p53 is a tumor-suppressor protein that interrupts the cell cycle to permit DNA repair [20]. The mutation and subsequent inactivation of p53 mark a critical early pathogenic step in SCC development, as this dysfunction allows cells to evade apoptosis, leading to the clonal proliferation of the mutated cells [21]. Multiple studies have documented the overexpression of p53 in precancerous skin lesions and SCC, as well as in skin after sun exposure, with elevated p53 levels recognized as a biomarker of recent sun exposure [22]. Increased p53 has been associated with progression from AK to SCC [13].

p63, a member of the p53 family of transcription factors, plays a pivotal role in keratinocyte differentiation across various stages. While p63 is abundantly expressed in the basal and suprabasal layers of normal epidermis, its expression diminishes in the upper spinous layer and is absent in the granular and cornified layers. Aberrant p63 expression is a hallmark of SCC across different organ sites, contributing to oncogenesis by disrupting numerous cell processes, including the inhibition of oncogene-induced keratinocyte senescence and reducing the function of other p53 family members [23,24].

Cyclin D1 plays an important role for the G1-S phase transition within the cell cycle. It activates CDK4 and CDK6, leading to the phosphorylation of the retinoblastoma protein, which in turn initiates transcription and activates proteins involved in the G1 checkpoint passage and S phase entry. Cyclin D1 overexpression truncates the G1 phase resulting in abnormal proliferation [25]. In situ SCCs have demonstrated more prevalent diffuse immunostaining of cyclin D1 compared to AK, suggesting a role in AK progression [26].

The p16 protein, encoded by the CDKN2A gene, impedes the cell cycle’s progression from the G1 to S phase by binding to and inactivating CDK4 and CDK6. It interacts with retinoblastoma and p53 tumor-suppressor genes in its role as a cell cycle inhibitor [27]. UV radiation activates p16, and p16-expressing cells have been detected in AK lesions. In a mouse model, persistent p16 expression in a subset of epidermal cells was shown to induce hyperplasia and dysplasia, promoting tumor formation after mutagenesis [16].

Ki67, a nuclear antigen, is a marker of cell proliferation across various cycle phases (G1, S, G2, M) and is absent in the quiescent phase (G0). Ki67 expression correlates with mitotic count and is employed as a surrogate marker for assessing the rate of proliferation, which is crucial for the diagnosis, classification, and prognosis of various neoplasms. SCC arising from AK through the differentiated pathway exhibits significantly higher Ki67 scores compared to the classical pathway. This increased proliferation also explains the progression of AK from a lesion with basal atypia into a bowenoid neoplasia with complete replacement of the epidermis by atypical cells and frequent mitoses in all epithelial layers [28]. 

The BCL-2 gene (B-cell lymphoma-2) encodes a protein that inhibits apoptosis. Within normal skin, Bcl-2-positive basal cells serve as a reservoir for the renewal of squamous epithelium [29]. In sun-exposed skin, Bcl-2 prevents UV-induced cell death [30]. An upregulation of this anti-apoptotic Bcl-2 protein is implicated in the progression from AK to SCC [12]. Additionally, Bcl-2 can label inflammatory cell infiltrate [31] and, within the complex process of cancer development, the extent of dermal infiltrate correlates significantly with keratinocyte atypia, mitotic count, and adnexal involvement [32]. 

CD31 is recognized as a highly sensitive and specific endothelial marker in paraffin tissue samples [33]. Neovascularisation plays a critical role in tumor development and progression. Studies have previously demonstrated a significant increase in the microvascular area corresponding with the transition from AK to cutaneous SCC [34]. 

In this cross-sectional study, we compared the expression of immunohistochemical markers—p53, p63, p16, Ki67, Cyclin D, Bcl-2, and CD31—between a group presenting with intraepithelial keratinocyte neoplasia, specifically AK and SCC in situ (experimental group), and a control group with normal-appearing skin. Our objective was to discern which marker associations are unique to the disease and which can be anticipated in elderly skin exposed to sunlight. To our knowledge, this study is the first to evaluate these seven immunohistochemical markers concurrently in AK and SCC in situ, and healthy sun-exposed skin. The findings derived from this study are anticipated to lay the groundwork for subsequent investigations into the relationship between immunohistochemical marker expression and clinical as well as dermatoscopic characteristics.

## 2. Materials and Methods

### 2.1. Patient Population

Considering the increased prevalence of AK and SCC among the aging population, our study included elderly fair-skinned individuals. These patients underwent examination by a board-certified dermatologist who diagnosed and selected the biopsy sites for AK or SCC in situ. The extent of sun damage was quantified using the AK field assessment scale [35] according to which all participants exhibited moderate to severe photodamage, necessitating more frequent follow-up consultations. Evaluated parameters included the count of AK lesions, evidence of field cancerization, wrinkles, pigmentation changes, telangiectasia, and cutaneous atrophy. Our patient sample did not have significant occupational sun exposure. However, given the prevalence of outdoor activities in the Latvian population and the low use of sunscreen among the elderly cohort, incidental sun exposure was considered noteworthy. 

### 2.2. Tissue Samples

A 3 to 4 mm punch biopsy was taken from lesions clinically and dermatoscopically diagnosed as AK or SCC in situ. These samples underwent routine histopathological evaluation and were included in the study following confirmation of the diagnosis by a study-unrelated pathologist. Control biopsies were prospectively collected from the maxillofacial surgery department specifically from lateral margins of elliptical excisions or remnant tissues from flap preparations. In total, the following number of biopsies were included: 22 from AK, 7 from SCC in situ, and 8 control biopsies. All patients from whom samples were obtained were identified as Caucasian, and all biopsies originated from facial tissue.

### 2.3. Histopathological and Immunohistochemical Analyses

The extent of atypical keratinocytes within the epidermis was stratified into three grades, as delineated by Röwert-Huber et al. [4]. Similarly, the basal growth pattern was assessed and categorized into three grades (protruding I–III) following the criteria established by Schmitz et al. [36].

Immunohistochemical staining was carried out on formalin-fixed paraffin-embedded tissue using both a DAKO Autostainer Plus (DAKO, Glostrup, Denmark) and Ventana Benchmark XT automated immunostainer (Ventana Medical System, Tuscon, AZ, USA). Detailed information regarding the monoclonal antibodies, clones, dilutions, incubation times, and manufacturers is presented in Table 1. All staining procedures included the use of appropriate controls.

Nuclear staining for p16, p53, and p63 was deemed positive, regardless of any associated cytoplasmic staining. The expression levels of p16 and p53 were semi-quantitatively assessed using predefined thresholds: <1%, 1–30%, 30–50%, and >50% for p16 and weak staining (if any), <5%, 5–50%, and >50% for p53. The epidermal distribution of p53, p63, Ki67, CyclinD, and Bcl-2 staining was categorized similarly to the extent of cellular atypia observed in hematoxylin and eosin-stained sections, with five distinct groups: no expression, expression in the lower third, expression up to the lower two-thirds, expression above two-thirds, and full-thickness expression. CD31 expression was quantified by manually counting positively stained vessels across three 20× magnification fields and semi-quantitatively classified into three categories: isolated, scattered peri-vascular inflammatory cells; foci of moderately pronounced inflammation; and areas of pronounced inflammation.

### 2.4. Statistical Analysis

The independent sample T-test was employed to evaluate the age difference between the experimental and control groups, predicated on the data’s normal distribution, verified by the Shapiro–Wilk test and normal Q-Q plots, and homogeneity was confirmed by Levene’s test. Fisher’s exact test was performed to investigate the association between the patterns of expression. The *p* < 0.05 was considered statistically significant. All statistical analyses were performed using Jamovi Version 2.3.28, retrieved from https://www.jamovi.org, accessed on 11 July 2023.

### 2.5. Ethics

All patients provided their informed written consent before the biopsies were taken. The study conformed to the ethical standards of the Declaration of Helsinki and received approval from the relevant local ethics committees.

## 3. Results

Table 2 summarizes the demographic, pathological, and immunohistochemical characteristics. Mean age exceeded 73 years with no significant difference between experimental and control groups (*p* > 0.05). Both groups had a female predominance. 

Sunscreen use was evaluated within the patient cohort. Among female participants, 27.3% reported consistently using sunscreen on sunny days. Conversely, all male participants denied any sunscreen use. However, this observed difference in sunscreen usage between genders was not statistically significant.

As expected, our findings revealed that there are statistically significant differences (*p* < 0.05) in the presence and extent of cellular dysplasia, as well as the degree of protruding, between AK/SCC samples and control biopsies. These differences are markers for distinguishing AK/SCC lesions.

Furthermore, statistically significant differences (*p* < 0.05) were observed between the experimental and control groups in various evaluations, including the presence of amorphous masses, the intensity of p53, the extent of Bcl-2 in epidermal layers, quantitative expression of Bcl-2 in the epidermis, subepidermal infiltration of Bcl-2, and the amount of CD31 dermal infiltrate. Representative images of the expression of immunohistochemical markers can be found in Figure 1 and Figure 2.

## 4. Discussion

Exposure to ultraviolet (UV) light is the primary environmental factor responsible for the onset and progression of skin cancer. Consequently, areas of the skin that are frequently exposed to sunlight, such as the face, are the most common sites for the development of skin cancer [5,37,38].

In this study, we conducted a comparative analysis of histopathological and immunohistochemical markers between samples of normal skin and intraepidermal keratinocyte neoplasia, specifically AK and in situ SCC, which were collected prospectively. Additionally, we evaluated the potential impact of age and UV exposure on the expression of immunohistochemical markers by focusing on a cohort of elderly patients [15,17,39]. The patient cohort in our study was predominantly female. This predominance is noteworthy given that AKs are typically more frequently diagnosed in men, who, due to higher rates of baldness, often have greater skin exposure to the sun. Nonetheless, our findings align with other studies that have similarly reported a higher incidence of AKs among women [40,41,42]. Furthermore, the predominance of women in our study sample may be attributed to the differences in life expectancy between genders in Latvia, where in 2021, it was 68.2 years for men and 77.9 years for women. This is particularly relevant given that the mean age of our patient cohort was 78.1 years [43]. Contrary to the common perception that women tend to use a broader range of cosmetic products, including sunscreens, more frequently than men, our study’s findings did not show a statistically significant gender difference in sunscreen use. However, the use of moisturizing creams exhibited a notable gender disparity, with 66.7% of women in our cohort using these products compared to none of the men.

In summary, our study identified differences in the prevalence of amorphous masses, the intensity of p53 staining, and the levels of Bcl-2 and CD31 expression. 

In the majority of AK and SCC cases, we identified amorphous masses, a finding not seen in any of the control biopsies. This observation aligns with findings from previous research, which reported a high incidence of severe solar elastosis in cutaneous SCC [44,45]. Severe solar elastosis, marked by the substitution of dermal collagen with elastotic material, results from prolonged and excessive UV damage [44].

The abnormal expression of p53 is considered a marker of premalignant lesions and plays a central role in the development of SCC [20]. A notable increase in p53 staining correlates with aging and exposure to sunlight, while its expression diminishes with the consistent application of sunscreen [37]. Essentially, progression from AK to SCC is associated with increased p53 staining [13,15,46]. Simultaneously, research by Piipponen et al. identified a fraction of AK, in situ SCC (8%), and cutaneous SCC (11%) lesions that were p53 negative, suggesting a possible nonsense mutation in the tumor protein 53 gene leading to the absence of functional p53 protein [47]. Similarly, Neto et al. found a higher incidence of SCC adjacent to AK lesions where less than 25% of cells were p53 positive [20]. In our study, we saw a slightly higher intensity of p53 staining in the AK/SCC cohort, aligning with the literature, while no notable differences were found in the pattern or extent of p53 expression across the epidermal layers between the two groups. Similarly, as observed by Neto et al. [20] and Piipponen et al. [47], not all AK cases in our study showed p53 expression. More than 5% immunoreactivity to p53 was observed in 75.8% of the AK/SCC samples and the staining was considered strong in 44.8% of the cases. 

Bcl-2 is an antiapoptotic molecule located in the mitochondrial membrane. Alteration of its function can promote cancer development. Prior research has noted an increase in Bcl-2 expression in AK [48]. Additionally, studies have indicated a significantly higher prevalence of Bcl-2-positive tumor cells in SCC compared to asymptomatic AK. Furthermore, a stepwise increase in Bcl-2 expression has been documented from asymptomatic AK to inflamed AK and subsequently to SCC, suggesting a pathway of progression through inflamed AK [46]. Our results agree with previous studies, showing higher expression of Bcl-2 in the epidermis of both AK and in situ SCC samples. Moreover, Bcl-2 stained dermal inflammatory cells and inflammation is associated with a progression of AK to SCC due to its role in generating reactive oxygen species and fostering immune responses, cellular transformation, survival, proliferation, invasion, angiogenesis, and metastasis [48,49,50]. Our results also revealed that both the Bcl-2 subepidermal infiltrate and CD31 expression, which is the most sensitive and specific endothelial marker in paraffin-embedded sections [33], are elevated in the AK/SCC samples compared to controls. 

One of the strengths of this study is that all the biopsies were consistently taken from the facial area. The results obtained allow us to further analyze associations between dermatoscopical features and p53, CD31, and Bcl-2 expression within the same cohort and avoid analyzing markers that showed no difference with the control group. Such insights have the potential to expand our understanding of the interplay between dermatoscopic characteristics and immunohistochemical marker expression. This could also potentially enhance the diagnostic precision of dermoscopy by pinpointing features that correlate with more aggressive histological and immunohistochemical profiles, such as adnexal involvement, protruding, and increased expression of p53, CD31, and Bcl-2.

## 5. Conclusions

The presence of amorphous masses and the expression of p53, Bcl-2, and CD31 differ between sun-exposed skin and cutaneous squamous cell carcinomas in situ, including actinic keratoses.

## Figures and Tables

**Figure 1 jcm-12-07291-f001:**
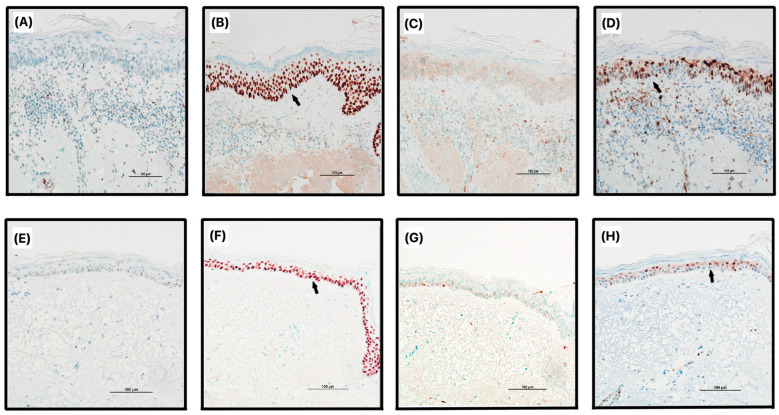
Immunohistochemical expression of p53, p63, p16, and Cyclin D in actinic keratosis (**A**–**D**) and control (**E**–**H**) skin. Black arrows indicate strong positive cell staining. (**A**) expression of p53 in actinic keratosis (AK) with weak staining if any; (**B**) expression of p63 in AK, distribution above 2/3 of epidermal thickness; (**C**) expression of p16 in AK with 1–30% nuclear staining; (**D**) expression of Cyclin D in AK, distribution up to 2/3 of epidermal thickness; (**E**) expression of p53 in control skin (CS) with <5% nuclear staining; (**F**) expression of p63 in CS, distribution above 2/3 of epidermal thickness; (**G**) expression of p16 in CS with <1% nuclear staining; (**H**) expression of Cyclin D in CS, distribution up to 2/3 of epidermal thickness.

**Figure 2 jcm-12-07291-f002:**
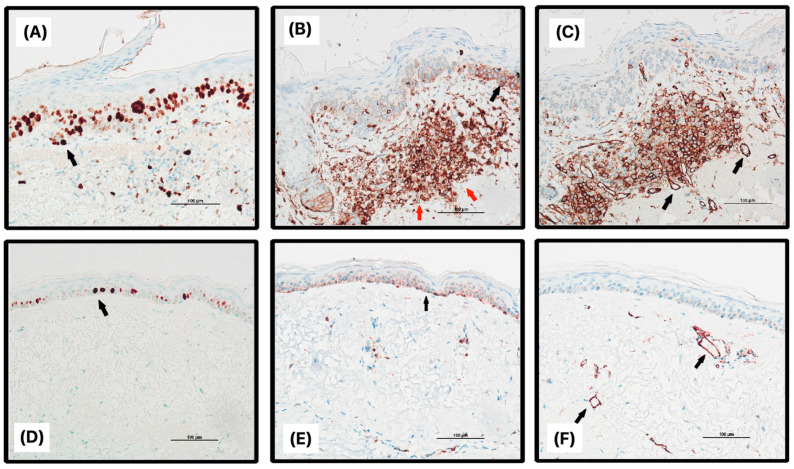
Immunohistochemical expression of Ki67, Bcl-2, and CD31 in actinic keratosis (**A**–**C**) and control (**D**–**F**) skin; (**A**) expression of Ki67 in actinic keratosis (AK), distribution up to 2/3 of epidermal thickness; (**B**) expression of Bcl-2 in AK, distribution up to 2/3 of epidermal thickness, foci with pronounced subepidermal infiltration (red arrows); (**C**) expression of CD31 in AK, visible pronounced expression; (**D**) expression of Ki67 in control skin (CS), distribution of up to 1/3 of epidermal thickness; (**E**) expression of Bcl-2 in CS, distribution up to 2/3 of epidermal thickness, weak or none subepidermal infiltration; (**F**) expression of CD31 in CS, few visible positive capillaries. Black arrows indicate positive cell staining in the epidermis (**A**,**B**,**D**,**E**) and positive endothelial staining (**C**,**F**).

**Table 1 jcm-12-07291-t001:** List of antibodies and detection methods used in the study.

Antibody	Clone	Dilution	Incubation Time	Source
p53	Do-7	Ready-to-use	30 min	DAKO
p63	A4A	Ready-to-use	20 min	Ventana
p16	E6H4	Ready-to-use	20 min	Ventana
Ki67	MIB1	Ready-to-use	16 min	DAKO
Cyclin D	EP12	Ready-to-use	30 min	DAKO
Bcl-2	124	Ready-to-use	30 min	DAKO
CD31	JC70A	Ready-to-use	30 min	DAKO

**Table 2 jcm-12-07291-t002:** Distribution of histopathological and immunohistochemical signs of biopsied lesions.

	Cases, Total (%)	Controls, Total (%)	*p* Value
Total number of cases included	29 (100%)	8 (100%)	
Mean age (±SD)	78.1 (±5.8) years	73.8 (±6.3) years	*p* = 0.076
Gender:			
Males	7 (24.1%)	3 (37.5%)	*p* = 0.655
Females	22 (75.9%)	5 (62.5%)	
Extent of atypical keratinocytes in the epidermis [4]			
No atypical keratinocytes	0 (0%)	8 (100%)	*p* < 0.001 *
In the lower 1/3	4 (13.8%)	0 (0%)	
Up to lower 2/3	14 (48.3%)	0 (0%)	
Above 2/3	4 (13.8%)	0 (0%)	
Full thickness	7 (24.1%)	0 (0%)	
Protruding [36]			
None	5 (17.2%)	8 (100%)	*p* < 0.001 *
Crowding (pro I)	7 (24.1%)	0 (0%)	
Budding (pro II)	3 (10.3%)	0 (0%)	
Papillary sprouting (pro III)	14 (48.3%)	0 (0%)	
Presence of amorphic masses	21 (72.4%)	0 (0%)	*p* < 0.001 *
p16 semiquantitative expression			
<1%	13 (44.8%)	4 (50%)	*p* = 0.865
1–30%	10 (34.5%)	4 (50%)	
30–50%	3 (10.3%)	0 (0%)	
>50%	3 (10.3%)	0 (0%)	
p16 distribution			
Negative	13 (44.8%)	4 (50%)	*p* = 1.0
Patchy	13 (44.8%)	4 (50%)	
Patchy/Diffuse	3 (10.3%)	0 (0%)	
p53 semiquantitative expression			
Weak staining (if any)	5 (17.2%)	1 (12.5%)	*p* = 0.740
<5%	2 (6.9%)	0 (0%)	
5–50%	11 (37.9%)	5 (62.5%)	
>50%	11 (37.9%)	2 (25.0%)	
p53 distribution in the epidermis			
No expression	6 (20.7%)	1 (12.5%)	*p* = 0.885
In the lower 1/3	9 (31.0%)	3 (37.5%)	
Up to lower 2/3	4 (13.8%)	2 (25.0%)	
Above 2/3	9 (31.0%)	2 (25.0%)	
Full thickness	1 (3.4%)	0 (0%)	
p53 staining intensity			
No staining	6 (20.7%)	0 (0%)	*p* = 0.036 *
Weak staining	10 (34.5%)	7 (87.5%)	
Strong staining	13 (44.8%)	1 (12.5%)	
p63 distribution in the epidermis			
Up to lower 2/3	5 (17.2%)	1 (12.5%)	*p* = 1.0
Above 2/3	23 (79.3%)	7 (87.5%)	
Full thickness	1 (3.4%)	0 (0%)	
Cyclin D distribution in the epidermis			
In the lower 1/3	11 (37.9%)	0 (0%)	*p* = 0.075
Up to lower 2/3	14 (48.3%)	7 (87.5%)	
Above 2/3	4 (13.8%)	1 (12.5%)	
Ki67 distribution in the epidermis			
In the lower 1/3	14 (48.3%)	7 (87.5%)	*p* = 0.398
Up to lower 2/3	11 (37.9%)	1 (12.5%)	
Above 2/3	2 (6.9%)	0 (0%)	
Full thickness	2 (6.9%)	0 (0%)	
Bcl-2 distribution in the epidermis			
No expression	16 (55.2%)	0 (0%)	*p* = 0.014 *
In the lower 1/3	8 (27.6%)	5 (62.5%)	
Up to lower 2/3	4 (13.8%)	2 (25.0%)	
Above 2/3	1 (3.4%)	1 (12.5%)	
Bcl-2 semiquantitative expression in the epidermis			
No or almost no cells	16 (55.2%)	0 (0%)	*p* = 0.015 *
Weak (<10%)	6 (20.7%)	4 (50.0%)	
Moderately positive (10–25%)	3 (10.3%)	2 (25.0%)	
Highly positive (>25%)	4 (13.8%)	2 (25.0%)	
Bcl-2 subepidermal infiltration			
Weak or none	2 (6.9%)	7 (87.5%)	*p* < 0.001 *
Foci with more pronounced	22 (75.9%)	1 (12.5%)	
Foci with very pronounced	5 (17.2%)	0 (0%)	
CD31 expression in dermal capillaries			
Up to 20 positive capillaries at 20× magnification	1 (3.4%)	2 (25.0%)	*p* = 0.174
20–40 positive capillaries at 20× magnification	19 (65.5%)	5 (62.5%)	
Above 40 positive capillaries at 20× magnification	9 (31.0%)	1 (12.5%)	
CD31 semiquantitative dermal expression			
Separate scattered positive vessels	2 (6.9%)	3 (37.5%)	*p* = 0.011 *
Foci with moderately pronounced expression	8 (27.6%)	5 (62.5%)	
Foci with pronounced expression	12 (54.5%)	0 (0%)	

* *p* value < 0.05 was considered statistically significant.

## Data Availability

Data is available upon reasonable request. The data are not publicly available due to the continuous nature of the study.

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
