# Peer review of "Expression of p53, p63, p16, Ki67, Cyclin D, Bcl-2, and CD31 Markers in Actinic Keratosis, In Situ Squamous Cell Carcinoma and Normal Sun-Exposed Skin of Elderly Patients"

_jcm, 2023, doi:10.3390/jcm12237291_

Round 1
Reviewer 1 Report
Comments and Suggestions for Authors
Dear Authors,
The paper is very interesting and original. The paper is well structured, matherials and methods are well described and the results are rich. It would be interesting to add how these immunohistochemical evaluations could be correlated with dermatoscopy or confocal microscopy and what advantages this would bring to the patient.
Thanks very much
Best regards
Author Response
Dear Reviewer,
Thank you for your suggestion. We have incorporated additional information in response to your input. A more comprehensive exploration of this topic is planned for our subsequent study.
Reviewer 2 Report
Comments and Suggestions for Authors
A topic studied by Latvian researchers is interesting one, well performed and written. However it still requires some discussion and correction as follows.
1.. I would expect more information concerning patients. Do the head a regular sun(UV) light exposure or suffer intensive exposure as let say fisherman or field workers?
2. An abundance of female in the study group requires more comment at least because o more prevalent usage of cosmetics including skin protection media.
3. Analysis of proteins was performed only on semi-quantitative way (mentioned in the text) that would be not sufficient for more sophisticated study.
4. Ethical rules have been observed.
5. The list of references require more attention as
- a majority a journal titles are given in abbreviations (as recommended) some (e.g. [2], [17], [21], [29]) are not,
- italics were applied to give journal titles except[ 2]9,
- position [7] is not having full bibliographic information,
Recommendation: minor revision
Author Response
Dear Reviewer,
We are grateful for your suggestions and the time dedicated to reviewing our manuscript.
- We have improved both the Methods and Discussion sections with additional data. Notably, we have included a new paragraph acknowledging that, while our patient sample lacked significant occupational sun exposure, the general propensity for outdoor activities among the Latvian population (especially gardening), coupled with the minimal use of sunscreen in our elderly cohort, renders incidental sun exposure a significant factor.
- We have also included remarks on the use of sunscreens and cosmetic products.
- We agree with your comment regarding a more sophisticated evaluation of the markers and we intend to implement this in our future work for the markers we found differently expressed.
- Lastly, we appreciate your attention to the issues within the references and have addressed these accordingly."
Reviewer 3 Report
Comments and Suggestions for Authors
In the current research article, authors have explored "Expression of p53, p63, p16, Ki67, Cyclin D, Bcl-2, and CD31 markers in actinic keratosis, in situ squamous cell carcinoma and normal sun-exposed skin of elderly patients". The subject is of interest and falls in the topics of “Journal of Clinical Medicine” Journal. The study needs some corrections.
After reviewing the manuscript thoroughly, I have following comments:
The English language needs to improve throughout the manuscript. There are number of grammatical and spelling mistakes in the manuscript. Check throughout the manuscript.
Abstract should be improved.
What is prospective study?
What is reason behind to select elderly patients?
Novelty of the study must be added in the introduction part of manuscript.
Clearly indicate the role of these markers p53, p63, p16, Ki67, Cyclin D, Bcl-2, and CD31 in actinic keratosis in introduction section. Elaborate in situ squamous cell carcinoma in introduction part as well.
Check the abbreviations throughout the manuscript and insert the where required.
In figure 1 and 2, indicate the major changes using labels with arrow heads for better clearance.
Comments on the Quality of English LanguageThe English language needs to improve throughout the manuscript. There are number of grammatical and spelling mistakes in the manuscript. Check throughout the manuscript.
Author Response
Dear Reviewer,
We are grateful for your suggestions and the time dedicated to reviewing our manuscript.
- Due to the word limit for the abstract, we could only make a few small edits. We added more details in other parts of the paper to clarify things.
- We agree that “prospective” should have been formulated better. The patients were collected prospectively. The study is cross-sectional.
- We included elderly patients as the prevalence of actinic keratosis and invasive squamous cell carcinoma increases with age. Besides we performed the biopsy for patients whose lesion characteristics would benefit from biopsy. An explanation of this has been added to the manuscript.
- We’ve also expanded the introduction to include elaboration of in situ squamous cell carcinoma, to indicate the roles of markers and the novelty.
- Arrows have been added to the Figures.